# VTruST : Controllable value function based subset selection for Data-Centric Trustworthy AI

## Abstract

Trustworthy AI is crucial to the widespread adoption of AI in high-stakes applications with *explainability, fairness and robustness* being some of the key trustworthiness metrics. *Data-Centric AI* (DCAI) aims to construct high quality datasets for efficient training of trustworthy models. In this work, we propose a controllable framework for data-centric trustworthy AI (DCTAI)- VTruST, that allows users to control the trade-offs between the different trustworthiness metrics of the constructed training datasets. A key challenge in implementing an efficient DCTAI framework is to design an online value-function-based training data subset selection algorithm. We pose the training data valuation and subset selection problem as an online sparse approximation formulation, where the features for each training datapoint is obtained in an online manner through an iterative training algorithm. We propose a novel online-version of the OMP algorithm for solving this problem. We also derive conditions on the data matrix, that guarantee the exact recovery of the sparse solution. We demonstrate the generality and effectiveness of our approach by designing data-driven value functions for the above trustworthiness metrics. Experimental results show that VTruST outperforms the state-of-the-art baselines for fair learning as well as robust training, on standard fair and robust datasets. We also demonstrate that VTruST can provide effective tradeoffs between different trustworthiness metrics through pareto optimal fronts. Finally, we show that the data valuation generated by VTruST can provide effective data-centric explanations for different trustworthiness metrics.

## 1 Introduction

The field of artificial intelligence (AI) has seen rapid advancements leading to its involvement in our daily lives. This usefulness comes with heightened responsibility especially in high-stake applications such as predicting the risk of criminal recidivism (Angwin et al., 2016), autonomous driving (Kohli & Chadha, 2020; Chen et al., 2021), disaster management (Linardos et al., 2022) and many more. In order for users to trust the decision of the AI algorithms, it is essential for them to be fair, reliable and robust under different circumstances. There is a substantial body of research in the area of model-centric AI where the need for making the models robust (Wang et al., 2022; Hataya & Nakayama, 2022) or fair (Roh et al., 2020; Romano et al., 2020) or unbiased (Gat et al., 2020; Majumdar et al., 2021) or explainable (Pillai & Pirsiavash, 2021; Sarkar et al., 2022) has been explored while keeping the data fixed. However, the core input to the models, *the data*, has undergone reasonably less exploration. With the upcoming progress in the field of *Data-Centric AI* [1] (Hajij et al., 2021; Motamedi et al., 2021), currently the focus has shifted to creating quality datasets to avoid data cascades in real world setting [2]. The DCAI approach comes with several advantages - identifying data points that explain certain model characteristics (Seedat et al., 2022b; Ethayarajh et al., 2022), selecting useful datapoints for efficient training (Das et al., 2021a; Pooladzandi et al., 2022), discarding noisy samples deviating training trajectory (Pruthi et al., 2020; Das et al., 2021b),

---

[1] https://datacentricai.org/
[2] https://ai.googleblog.com/2021/06/data-cascades-in-machine-learning.html

scoring datapoints based on a value function reflecting their contribution to the task (Yoon et al., 2020; Lin et al., 2022) and many more.

In this work, we design a data-centric approach to achieving trustworthy AI through a controllable data valuation framework with the composition of several value functions aimed at key trustworthy AI metrics - fairness, robustness, and explainability. Additionally, we allow the users to control the balance between the trustworthiness metrics. We pose the problem of training data valuation and selection as an online sparse approximation problem, which we execute using a novel online version of OMP (Cai & Wang, 2011). Unlike in OMP where the decision of including the optimal variables happens iteratively *looking at the entire data each time*, our proposed method, VTruST, works in an online setup where the data arrives incrementally and the decision of inclusion/exclusion happens on the fly by *only iterating over the already included variables*. We add a *DataReplace* component which enforces conditions based on the projections leading to the replacement of already selected non-optimal features. To the best of our knowledge, this is one of the first works to provide a single unified framework for balancing between the multiple value functions, followed by data valuation and extraction of subsets. These subsets when trained from scratch, lead to fairer, more robust, or more accurate models depending on the usage of value functions.

To summarize, our main contributions in the paper are: (1) We design a data-centric controllable selection framework that helps us obtain a subset of training datapoints specifically suited for trustworthiness metrics. (2) We formulate the value functions for the metrics of fairness, robustness, and accuracy. We also design a sampling algorithm for constructing augmented datasets for robustness. (3) We provide theoretical derivations of the conditions used in VTruST and show that they are sufficient for the exact recovery of the sparse solution. (4) We demonstrate that VTruST with subsets of even 20% of the original dataset, is able to outperform all state-of-the-art baselines by $\sim 10 - 20\%$ in both the tasks of fairness and robustness and can also provide data-centric explanations behind its performance.

## 2 VTruST: Value-driven Trustworthy AI through Selection of Training Data

We propose a controllable value function-based framework for developing trustworthy models using a data-centric paradigm. Our system has two main components: (1) A general value function-based framework that allows users to specify a combination of trustworthiness metrics, and (2) a novel online subset selection algorithm for constructing high-quality training dataset based on the specified value function. Section 2.1 describes the framework under which different value functions can be specified for different trustworthiness metrics. Section 2.2 defines different value functions for data-centric trustworthy AI. Section 2.3 describes the sparse approximation-based online subset selection algorithm that efficiently selects subset which can optimally approximate the value function.

### 2.1 A Controllable Value Function-based Framework for DCTAI

Let $\mathcal{D} = \{d_i = (u_i, w_i) | i = 1, 2, .., N\}$ be the training dataset for a supervised learning task with input $u_i \in \mathcal{U}$ and labels $w_i \in \mathcal{W}$. Similarly, let $\mathcal{D}' = \{d'_j = (u'_j, w'_j) | j = 1, 2, .., M\}$ be the validation dataset which is used for assigning values/scores to a training datapoint $d_i \in \mathcal{D}$ based on a value function. The total value function, $\mathcal{V}^T(d'_j) = \sum_{t=1}^{T} \sum_{i=1}^{N} v_i^t(d'_j) \ \forall d'_j \in \mathcal{D}'$, defines a measure of the utility of the training dataset, defined for each datapoint validation dataset $\mathcal{D}'$. We also overload the notation to define the value function vector $\mathcal{V}^T(\mathcal{D}') = \sum_{t=1}^{T} \sum_{i=1}^{N} v_i^t(\mathcal{D}')$. The total value function is defined as an additive function over $v_i^t(\mathcal{D}')$, the value contributed by the training datapoints $d_i$ on the $t^{th}$ epoch, $t = 1, ..., T$. A commonly used value function in the data valuation literature (Pruthi et al., 2020), which is defined as the decrease in loss incurred due to an SGD update using the datapoint $d_i$: $v_i^t(d'_j) = l(\theta_t^{i-1}, d'_j) - l(\theta_t^i, d'_j)$, where $\theta_t^{i-1}$ and $\theta_t^i$ are the model parameters before and after the SGD update involving the training datapoint $d_i$ in the $t^{th}$ epoch. We call this the value function for accuracy, since each term measures the contribution of the training datapoint $d_i$ towards improving the validation loss which is a proxy for the validation accuracy: $\mathcal{V}_a(d'_j) = \sum_{t=1}^{T} \sum_{i=1}^{N} v_i^t(d'_j) = \sum_{t=1}^{T} \sum_{i=1}^{N} (l(\theta_t^{i-1}, d'_j) - l(\theta_t^i, d'_j))$. Other value functions for fairness and robustness are defined in the next section.

Given any value function $\mathcal{V}$, our aim is to find a subset of training datapoints $S \in \mathcal{D}$ that lead to an accurate approximation of the total value function in each epoch, as a weighted summation of contributions from the selected set of points $S$. Let $y_t(d'_j) = \mathcal{V}^t(d'_j) = \sum_{k=1}^{t} \sum_{i=1}^{N} v_i^k(d'_j)$ be the cumulative value function till the $t^{th}$ epoch. We are interested in the following sparse approximation: $y_t(d'_j) \approx \sum_{d_i \in S \subseteq \mathcal{D}} \alpha_i [\sum_{k=1}^{t} v_i^k(d'_j)]$ , where $\alpha_i$ are the weights for each training datapoint $d_i$. Using a second order Taylor series expansion of the loss function $l(\theta_t^i, d'_j)$ around $l(\theta_t^{i-1}, d'_j)$, and plugging in the SGD update formula $\theta_t^i - \theta_t^{i-1} = \eta_t \nabla l(\theta_t^{i-1}, d_i)$, we obtain the following approximation for each term in the value function $l(\theta_t^i, \mathcal{D}') - l(\theta_t^{i-1}, \mathcal{D}') \approx \eta_t \nabla l(\theta_t^{i-1}, d_i)^T \nabla l(\theta_t^{i-1}, \mathcal{D}') + \mathcal{O}(||\theta_t^i - \theta_t^{i-1}||_2^2)$. We truncate the Taylor expansion till the second-order terms, and absorb the learning rates $\eta_t$ in the learned coefficients $\alpha_i$, in order to arrive at the following sparse approximation problem:

$$y_t(d'_j) \approx \sum_{d_i \in S} \alpha_i \sum_{k=1}^{t} \left[ X_i^k(d'_j) \right] \ \forall d'_j \in \mathcal{D}', \ t = 1, ..., T \tag{1}$$

where $X_i^k(d'_j) = \nabla l(\theta_k^{i-1}, d_i)^T \nabla l(\theta_k^{i-1}, d'_j) + \frac{(\nabla l(\theta_k^{i-1}, d_i)^T \nabla l(\theta_k^{i-1}, d'_j))^2}{2}$ are the features for the $i^{th}$ training point calculated in epoch $t$. We use the vectors $\vec{y_t} = [y_t(d'_j) | j \in \mathcal{D}']$ and $\vec{X_i^t} = [X_i^t(d'_j) | j \in \mathcal{D}']$ to denote the predictor and predicted variables over the entire validation set.

The main challenge in solving the above sparse approximation problem is that we need to store the cumulative features: $\sum_{k=1}^{t} X_i^k(d'_j)$, for all training and validation point-pairs $(i, j)$ and for all epochs $t = 1, ..., T$. This becomes computationally prohibitive for many practical scenarios. Instead, we solve the following *online sparse approximation* (OSA) problem for each epoch $t$:

$$y_t(d'_j) \approx \sum_{(p,q) \in S_t} \beta_p^q \left[ X_p^q(d'_j) \right] \ \forall d'_j \in \mathcal{D}', \ t = 1, ..., T \tag{2}$$

Here, $S_t$ is the set of selected training datapoints at the end of the $t^{th}$ epoch. Note that the set $S_t$ can contain features from any of the epochs $q = 1, ..., t$. We constrain the size of the selected training datapoints to be less than a user-specified parameter $k$. To summarize, we pose the online sparse approximation problem as: $\min_{S_1, S_2, .., S_T, \vec{\beta}} \sum_{t=1}^{T} || \sum_{(p,q) \in S_t} (\beta_p^q \vec{X}_p^q) - \vec{y_t} ||_2^2 \ s.t. |S_t| \leq k \ \forall t = 1, ..., T$, where $\vec{\beta}$ is the set of all learned coefficients $\beta_p^q$ for all $(p, q) \in S_t$. Note that at any point in time $t$, we only need to store the $\vec{\beta}$ and $S_t$ for only the current time instance. In section 2.3, we describe an online algorithm for solving the above problem. We also highlight that the value function $\mathcal{V}(d'_j)$ only needs to satisfy an additive property over the training datapoints and epochs in order for the above formulation to be valid. Hence, this framework is applicable to a composite value function $\mathcal{V}(d'_j) = \sum_f \lambda_f \mathcal{V}_f(d'_j)$, where each value function $\mathcal{V}_f(.)$ satisfies the additive property. This leads us to a general *controllable* framework for incorporating many trustworthiness value functions, controlled using the user-specified weights $\lambda_f$.

## 2.2 VALUE FUNCTIONS FOR TRUSTWORTHY DATA-CENTRIC AI

In this section, we define additive value functions aimed at the trustworthiness metrics - *robustness* and *fairness*.

**Robustness Value Function**: It was observed in (Rebuffi et al., 2021; Addepalli et al., 2022) that data augmentation improves robust accuracy and reduces robust overfitting. We use this idea of augmenting datapoints and define the *robust value function* $(\mathcal{V}_{rvf}(\mathcal{D}', \mathcal{D}'_a))$ as a composite value function over accuracy $(\mathcal{V}_a(\mathcal{D}'))$ and robustness$(\mathcal{V}_r(\mathcal{D}'_a))$. We define the accuracy value function $\mathcal{V}_a(\mathcal{D}') = \sum_{t=1}^{T} \sum_{d_i \in \{\mathcal{D} \cup \mathcal{D}_a\}} l(\theta_t^i, \mathcal{D}') - l(\theta_t^{i-1}, \mathcal{D}')$ using both validation dataset $\mathcal{D}'$ as described above. The robustness value function is defined as $\mathcal{V}_r(\mathcal{D}'_a) = \sum_{t=1}^{T} \sum_{d_i \in \{\mathcal{D} \cup \mathcal{D}_a\}} l(\theta_t^i, \mathcal{D}'_a) - l(\theta_t^{i-1}, \mathcal{D}'_a)$, where $\mathcal{D}_a$ and $\mathcal{D}'_a$ are the augmented data (formed by adding brightness, impulse noise, etc.) from $\mathcal{D}$ and $\mathcal{D}'$ respectively. Hence, the overall value function used for robustness experiments is $\mathcal{V}_{rvf}(\mathcal{D}', \mathcal{D}'_a) = \lambda \mathcal{V}_a(\mathcal{D}') + (1 - \lambda) \mathcal{V}_r(\mathcal{D}'_a)$ where $\lambda$ is a user-defined parameter controlling the tradeoffs between accuracy and robustness. We note that above value function can also be arrived at by simply defining the combined loss function: $l_c(\theta_t^i, \mathcal{D}', \mathcal{D}'_a) = \lambda l(\theta_t^i, \mathcal{D}') + (1 - \lambda) l(\theta_t^i, \mathcal{D}'_a)$,

and then defining the value function as usual: $\mathcal{V}_{rvf}(\mathcal{D}', \mathcal{D}'_a) = \sum_{t=1}^{T} \sum_{d_i \in \{\mathcal{D} \cup \mathcal{D}_a\}} l_c(\theta_t^i, \mathcal{D}', \mathcal{D}'_a) - l_c(\theta_t^{i-1}, \mathcal{D}', \mathcal{D}'_a)$. Hence, the above framework works directly in this setting. Note that, the obtained training subset is a mixture of both unaugmented and augmented data.

**Fairness Value Function**: Existing literature in fairness (Roh et al., 2020; Romano et al., 2020) uses equalized odds disparity and demographic parity disparity for achieving fair models. We use the *equalized odds disparity*-based (EOD) objective function defined in (Roh et al., 2020) as our compositional value function for fairness. Let $x \in \mathcal{X}$ be the input domain, $\{y_0, y_1\} \in \mathcal{Y}$ be the true binary labels, and $\{z_0, z_1\} \in \mathcal{Z}$ be the sensitive binary attributes. As before, we define the *fair value function* $\mathcal{V}_{fvf}(\mathcal{D}')$ as the compositional value function over accuracy ($\mathcal{V}_a(\mathcal{D}')$) and fairness ($\mathcal{V}_f(\mathcal{D}')$) i.e $\mathcal{V}_{fvf}(\mathcal{D}') = \lambda \mathcal{V}_a(\mathcal{D}') + (1 - \lambda)\mathcal{V}_f(\mathcal{D}')$. Analogous to other value functions, we define fairness value function as the change in EOD: $\mathcal{V}_f(\mathcal{D}') = \sum_{t=1}^{T} \sum_{d_i \in \mathcal{D}} ed(\theta_t^i, \mathcal{D}') - ed(\theta_t^{i-1}, \mathcal{D}')$. Here the EOD is defined as the maximum difference in accuracy between the sensitive groups ($z \in \mathcal{Z}$) pre-conditioned on the true label ($y \in \mathcal{Y}$): $ed(\theta, \mathcal{D}') = max(\|l(\theta, \mathcal{D}'_{y_0, z_0}) - l(\theta, \mathcal{D}'_{y_0, z_1})\|, \|l(\theta, \mathcal{D}'_{y_1, z_0}) - l(\theta, \mathcal{D}'_{y_1, z_1})\|)$ (Roh et al., 2021b). We note that contrary to the accuracy and robustness value functions, where there is one value for every validation datapoint, there is a single disparity measure for the entire validation dataset for fairness. As previously, we can re-define $\mathcal{V}_{fvf}(\mathcal{D}') = \sum_{t=1}^{T} \sum_{d_i \in \mathcal{D}} ld_c(\theta_t^i, \mathcal{D}') - ld_c(\theta_t^{i-1}, \mathcal{D}')$ where $ld_c(\theta, \mathcal{D}')$ is a combined loss function involving accuracy and disparity metric: $ld_c(\theta, \mathcal{D}') = \lambda.l(\theta, \mathcal{D}') + (1 - \lambda).ed(\theta, \mathcal{D}')$.

## 2.3 An online-OMP algorithm for online sparse approximation

In this section, we describe a novel online-OMP based algorithm for the *online sparse approximation problem*(OSA): $\min_{S_1, S_2, ..., S_T, \vec{\beta}} \sum_{t=1}^{T} \|\sum_{(p,q) \in S_t} (\beta_p^q \vec{X}_p^q) - \vec{y}_t\|_2^2 \quad s.t. |S_t| \leq k \ \forall t = 1, ..., T$. The key difference between OSA and standard sparse approximation setting is that in OSA, new columns $\vec{X}_p^q$ are added and the target value $\vec{y}_t$ is updated at each epoch $t$. Algorithm 1 describes the novel online OMP-based algorithm for OSA. Line 9 adds new columns $\vec{X}_i^t$ till the cardinality of $S_t$ reaches $k$. Once the buffer is saturated, the *DataReplace* module in Algorithm 2 is invoked in line 7 to potentially replace an existing selected column with the current new column. The criteria for replacement is to select the columns in the $S_t$ that contribute to a better approximation of the current value function $\vec{y}_t$. Hence a new column $\vec{X}_i^t$ gets selected if the current approximation error reduces after the replacement.We compute the projection of the current column, $\vec{X}_i^t$ and that of the selected columns $\vec{X}_q^p \ \forall p, q \in S_t$ on the existing residual vector $\vec{\rho}_t$, measured by $\pi$ and $\pi'$ respectively. We also denote by $\gamma$, the contribution of column $p, q \in S_t$ ($\beta_q^p$). The column $(p, q)$ in $S_t$ whose additive impact ($\pi' + \gamma$) is smaller than that of incoming colmun $(i, t)$, but larger than the current feature for replacement ($\vec{X}_q^p$), gets substituted with the incoming point in line 14 having conditioned on the projection and the beta quotient in line 8 of Algorithm 2.

Note that the proposed algorithm is different from OMP Cai & Wang (2011) since OMP scans the entire list of columns for improvement on approximation error till $k$ columns are selected. Hence it cannot be used in the online sparse approximation setting. A related work by Jain et al. (2011) proposes OMPR (OMP with replacement) which replaces columns using hard thresholding on the projections of columns on residuals. However, it is not applied in an online setting and the entire data is expected to be available during the approximation. Bayesian OMP Herzet & Dremeau (2014) unrealistically assumes knowledge of the distribution of columns. To the best of our knowledge, the current work is the first to attempt the problem of online sparse approximation. The per-epoch time complexity of OMP is $\mathcal{O}(kMN)$ and that of OMPR (Jain et al., 2011) $\mathcal{O}(\tau kMN)$ where $\tau$ is the total number of iterations for the convergence of their algorithm. The proposed method, *VTruST* has a total time complexity of $\mathcal{O}(kM(N - k))$.

**Conditions for Optimality of the Selected Subset**: The proposed algorithm solves the online sparse approximation problem, where the optimal selected subset depends on the order in which the columns are added to the problem. We derive the conditions for an ordering $\Pi(\vec{X})$ of columns of the global matrix $\vec{X}$ to produce an optimal set of columns $\vec{X}_{(T)}$, as the selected set using the VTruST algorithm. For simplicity, we follow the notation in Cai & Wang (2011). Let $\vec{X}$ be the set of all columns $\{\vec{X}_1, .., \vec{X}_N\}$ and $\vec{X}_{(T)}$ be an optimal set of columns approximating the value

function $\vec{y}$. Also, let $\vec{X}_{(O)}$ denote the set of columns not included in the optimal set $\vec{X}_{(T)}$. We index the columns corresponding to selected datapoints by $\vec{X}_{(s_m)}$ where $m \in \{1, 2, .., N\}$. Using $\vec{X}_{(T)}$, we can obtain the set of optimal selected columns as $\vec{X}_{(c_m)} : \vec{X}_{(s_m)} \cap \vec{X}_{(T)}$ ; and the non-optimal ones as $\vec{X}_{(o_m)} : \vec{X}_{(s_m)} \setminus \vec{X}_{(c_m)}$. The algorithm aims to reduce the residual $\vec{\rho}_m$ which as we know is $\vec{\rho}_m = \vec{y} - \vec{\beta}_{(s_m)} \vec{X}_{(s_m)}$. Using the above, we can also rewrite as: $\vec{\rho}_m = \vec{y} - \vec{\beta}_{(c_m)} \vec{X}_{(c_m)} - \vec{\beta}_{(o_m)} \vec{X}_{(o_m)}$. We define the theorem which states the conditions for selection of an optimal set of elements.

**Theorem 2.1 (Conditions in terms of Algorithm)** $\forall m \in \{k+1, \ldots N\} \; \exists \vec{X}_{(T)} \in \mathcal{X} \; s.t. \; \forall x \in \vec{X}_{(T)}$ , $\forall z \in \vec{X}_{(o_m)}$ *if* $|\vec{\rho}_m^T x| > |\vec{\rho}_m^T z|$ *and* $\beta_z < 0$ , *then x will be included in* $\vec{X}_{(s_{m+1})}$ *by replacing z that satisfies the condition* $max_z |\vec{\rho}_m^T z| + \beta_z$

The proof is provided in the Appendix.

---

**Algorithm 1 : VTruST**

1: **Input:**
    i. $k$ : Total number of datapoints to be selected
    ii. $\vec{y}$ : Targeted value function
    iii. $\vec{X}_i$ : Features of all training points $d_i \in \mathcal{D}$
    iv. $S$ : Set of selected datapoint indices
    v. $\vec{\beta} \in \mathbb{R}^{|S|}$: Weight of selected datapoints
2: **Initialize:**
    $S \longleftarrow \phi$ //Indices of selected datapoints
3: **for** each epoch $t \in \{1, 2, ..., T\}$ **do**
4:     **for** each datapoint $d_i \in \mathcal{D}$ **do**
5:         **Input:** $\vec{y}_t, X_i^t \quad \forall i \in \{1, 2, .., N\}, ||X_i^t||_2 = 1$
6:         **Process:**
7:         **if** $|S_{t-1}| = k$ **then**
8:             $S_t \longleftarrow$ **DataReplace**$(\vec{y}_t, \vec{\xi}_{t-1}, S_{t-1}, \vec{\beta}_{t-1}, \vec{X}_i^t)$
9:         **else**
10:            $S_t \longleftarrow S_{t-1} \cup \{i\}$
11:         **end if**
12:         Update $\vec{\beta}_t = \operatorname{argmin}_\beta ||\vec{y}_t - \sum_{p,q \in S_t} (\beta_q^p \vec{X}_q^p)||_2$
13:         Update $\vec{\xi}_t = \sum_{p,q \in S_t} \beta_q^p \vec{X}_q^p$
14:     **end for**
15: **end for**
16: **Output:**Final set of selected datapoint indices $S_T$, learned coefficients $\{\beta_q^p | p, q \in S_T\}$

**Algorithm 2 : DataReplace**$(\vec{y}_t, \vec{\xi}_t, S_t, \vec{\beta}_t, \vec{X}_i^t)$

1:   $\vec{\rho}_t = \vec{y}_t - \vec{\xi}_t$
2:   $\pi_{max} = -\infty$
3:   $(a, b) = \phi$
4:   $\pi \longleftarrow$ abs$(\vec{X}_i^t \cdot \vec{\rho}_t)$
5:   **for** each index $p, q \in S_t$ **do**
6:     $\pi' \longleftarrow$ abs$(\vec{X}_q^p \cdot \vec{\rho}_t)$
7:     $\gamma \longleftarrow \beta_q^p$
8:     **if** $\pi > \pi' \& \gamma \leq 0 \& (\pi' + \gamma) > \pi_{max}$ **then**
9:       $\pi_{max} \longleftarrow \pi' + \gamma$
10:       $a, b \longleftarrow p, q$
11:     **end if**
12:   **end for**
13:   **if** $(a, b) \neq \phi$ **then**
14:     $S \longleftarrow S \setminus \{a, b\} \cup \{t, i\}$
15:   **end if**
16:   return $S$

---

## 3 EXPERIMENTAL EVALUATION

In this section, we describe the datasets, models, and evaluation metrics used for the standard performance and trustworthiness metrics - Robustness and Fairness. We analyze the performance of *VTruST* using data-centric explanations of the subsets. All our experiments have been executed on a single Tesla V100 GPU.

### 3.1 ERROR RATE, FAIRNESS AND ROBUSTNESS ON TABULAR DATA

We evaluate VTruST against baselines (wholedata standard training (ST), random, SSFR (Roh et al., 2021a), FairMixup Mroueh et al. (2021) and FairDummies (Romano et al., 2020)) on four classification datasets: COMPAS (Angwin et al., 2016) , Adult Census (Kohavi et al., 1996), MEPS-20 (mep) and a synthetic dataset proposed by (Roh et al., 2021a). We use a 2-layer neural network for all the datasets. For evaluation, we use the error-rate (ER) and fairness measures of equalised odds (Hardt et al., 2016) and demographic parity (Feldman et al., 2015) following (Roh et al., 2021a).

**Fairness and Error Rate comparison (VTruST-F) with baselines**: We compare the performance metrics of VTruST-F with the baselines in Table 1. The better the model is, the lower its ER as well as its disparity/fairness measures. We can observe in Table 1 that VTruST-F with 60% selected subset outperforms all the other methods in terms of fairness measures by a margin of 0.01-0.10, and performs close to Wholedata-ST that yields the lowest ER. This denotes that it is able to condemn the error-fairness tradeoff to a certain extent, emerging out to be the best performing method. We report these results with standard deviation across 3 runs.

Table 1: **Comparison of VTruST-F with baselines over 60% subset for fairness evaluation.**

| Methods | Synthetic | | | COMPAS | | | AdultCensus | | | MEPS20 | | |
|---|---|---|---|---|---|---|---|---|---|---|---|---|
| | ER ±std | EO Disp ±std | DP Disp ±std | ER ±std | EO Disp ±std | DP Disp ±std | ER ±std | EO Disp ±std | DP Disp ±std | ER ±std | EO Disp ±std | DP Disp ±std |
| Wholedata-ST | 0.27 ±0.001 | 0.28 ±0.03 | 0.28 ±0.02 | 0.34 ±0.001 | 0.31 ±0.05 | 0.24 ±0.03 | 0.16 ±0.002 | 0.19 ±0.06 | 0.13 ±0.06 | 0.09 ± 0.001 | 0.09 ± 0.007 | 0.08 ± 0.0008 |
| Random | 0.29 ±0.001 | 0.24 ±0.02 | 0.26 ±0.02 | 0.35 ±0.002 | 0.20 ±0.10 | 0.23 ±0.09 | 0.19 ±0.002 | 0.16 ±0.05 | 0.13 ±0.05 | 0.12 ±0.017 | 0.06 ±0.02 | 0.08 ±0.005 |
| SSFR | 0.29 ±0.002 | 0.25 ±0.02 | 0.29 ±0.01 | 0.35 ±0.002 | 0.26 ±0.03 | 0.17 ±0.02 | 0.21 ±0.001 | 0.18 ±0.03 | 0.12 ±0.01 | 0.14 ±0.003 | 0.10 ±0.011 | 0.06 ±0.005 |
| Fair-Dummies | 0.29 ±0.002 | 0.22 ±0.02 | 0.22 ±0.01 | 0.35 ±0.002 | 0.24 ±0.02 | 0.17 ±0.01 | 0.16 ±0.002 | 0.14 ±0.01 | 0.10 ±0.01 | 0.12 ±0.001 | 0.13 ±0.005 | 0.08 ±0.003 |
| Fair-Mixup | 0.29 ±0.02 | 0.18 ±0.03 | 0.20 ±0.05 | 0.35 ±0.03 | 0.15 ±0.03 | 0.13 ±0.04 | 0.24 ±0.04 | 0.11 ±0.05 | 0.1 ±0.02 | 0.89 ±0.02 | 0.02 ±0.04 | 0.05 ±0.03 |
| VTruST-F | 0.28 ±0.001 | **0.17** **±0.01** | **0.19** **±0.01** | 0.34 ±0.002 | **0.15** **±0.01** | **0.13** **±0.01** | 0.18 ±0.001 | **0.11** **±0.03** | **0.05** **±0.01** | 0.09 ±0.003 | **0.01** **±0.001** | **0.05** **±0.0008** |

**Tradeoffs between Error rate, Fairness and Robustness (VTrust-F , VTruST-FR, VTrust-R)**: We observe the tradeoffs between error rate vs fairness (VTruST-F: Figure 1a), fairness vs robustness (VTruST-FR: Figure 1b and 1c), and error rate vs robustness (VTruST-R: Figure 1d) through pareto frontal curve. Figure 1 shows the pareto frontier for the different combinations. We perform the experiments of robustness on tabular data in 2 ways: (a) label flipping following Roh et al. (2021a) and (b) feature augmentation using SMOTE (Chawla et al., 2002). Error rate is measured on the clean test sets while robust error rates are measured on the flipped and augmented test sets.

For VTruST-F and VTruST-FR, we vary the weightage $\lambda \in \{0, 0.1, 0.3, 0.5, 0.7, 0.9, 1\}$ and for VTruST-R, we vary it between $0 - 1$ in steps of $0.5$. The factor $\lambda = 0$ puts the entire weightage on the disparity measure in the case of VTruST-F and VTruST-FR, and on robust error rate for VTruST-R. Figure 1 shows that VTruST-F, VTrust-FR, and VTruST-R follow the desired pattern across all the setups. We can observe that Wholedata-ST has a lower error rate but high disparity/robust error values. The other baselines continue to have a higher error rate and disparity compared to VTruST. *Note that we report the remaining results on other datasets in the Appendix.*

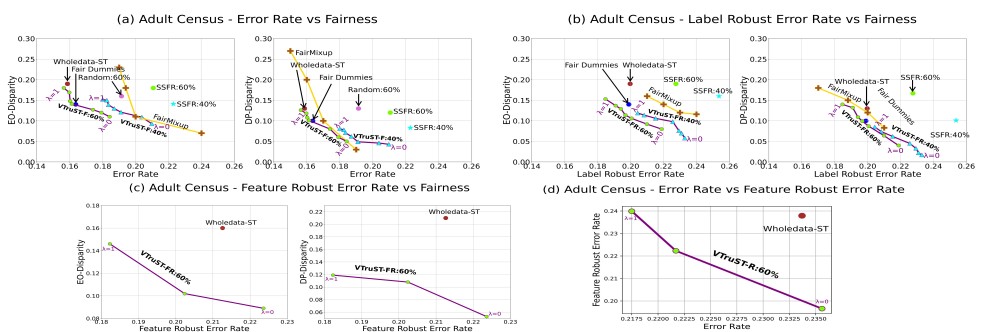

Figure 1: **Controlling tradeoffs in trustworthiness metrics in clean and augmented data setup for tabular domain.**

## 3.2 ACCURACY AND ROBUSTNESS ON IMAGE DATA

We evaluate VTruST on three image datasets: CIFAR10 (Krizhevsky et al., 2009) , MNIST (Deng, 2012) and Tinyimagenet (Le & Yang, 2015). We use ResNet-18 model (He et al., 2016) for all datasets. For evaluation, we use the standard accuracy (SA) computed on the clean test sets and the robust accuracy (RA) computed on their respective corrupted test sets, CIFAR-10-C, Tiny ImageNet-C (Hendrycks & Dietterich, 2019) and MNIST-C (Mu & Gilmer, 2019).

Despite the fact that augmentation leads to robustness (Rebuffi et al., 2021), it also leads to a large dataset with redundancy. Hence we use our framework to select a high-quality data for further evaluation. We adopt the data-centric approach in creating an augmented set followed by subset selection using VTruST. Next, we define the setup of the baselines.

(i) *Clean-ST*: Trained on unaugmented training set. (ii) *Uniform Augmentation (UAug)*: Trained on datasets added with augmentations chosen uniformly at random. (iii) *Sampled Augmentation (SAug)*: Trained on datasets added with augmentations sampled using Algorithm **??**. (Wang et al., 2021) (iv) *SSR* (Roh et al., 2021a): Trained on subset obtained using the objective function only for robustness. (v) *AugMax* (Wang et al., 2021).

Table 3: **Comparison of VTruST-R over varying subset sizes for robustness evaluation. The numbers in brackets indicate the difference with the second best among baselines.**

| Methods | MNIST | | | CIFAR10 | | | | TinyImagenet | | | |
|---|---|---|---|---|---|---|---|---|---|---|---|
| | #Data points | SA | RA | #Data points | SA | RA | Train Time (min/ep) | #Data points | SA | RA | Train Time (min/ep) |
| Clean-ST | 60K | 99.35 | 87.00 | 50K | 95.64 | 83.95 | 3.03 | 100K | 63.98 | 23.36 | 10.02 |
| AugMax | 240K | 97.62 | 88.79 | 200K | 94.74 | 86.44 | 13.78 | 400K | 54.82 | 40.98 | 61.2 |
| SAug | 260K | **99.36** (1.74) | **97.31** (8.52) | 200K | **94.9** (0.16) | **90.13** (3.69) | 12.1 | 300K | **62.04** (7.22) | **42.04** (1.06) | 28.7 |
| After subset selection from SAug | | | | | | | | | | | |
| SSR:20% | 52K | 98.79 | 86.51 | 40K | 92.17 | 79.29 | 2.45 | 60K | 20.26 | 16.05 | 10.35 |
| VTruST-R:20% | 52K | 98.76 | **92.46** (3.67) | 40K | 92.25 | 85.54 | 2.45 | 60K | 49.0 | 34.14 | 10.35 |
| SSR:40% | 104K | 98.98 | 94.96 | 80K | 93.3 | 85.73 | 4.9 | 120K | 32.82 | 24.42 | 13.5 |
| VTruST-R:40% | 104K | **99.04** (0.06) | **96.29** (1.33) | 80K | **94.74** | **88.23** (1.79) | 4.9 | 120K | **57.3** (2.48) | 39.69 | 13.5 |
| SSR:60% | 156K | 99.07 | 96.53 | 120K | 93.77 | 88.0 | 7.4 | 180K | 41.94 | 30.07 | 22.32 |
| VTruST-R:60% | 156K | **99.12** (0.05) | **97.09** (0.56) | 120K | **94.77** (0.03) | **89.21** (1.21) | 7.4 | 180K | **60.88** (6.03) | **41.50** (0.52) | 22.32 |

Table 4: **Performance comparison with data valuation methods on CIFAR10**

| Methods | GraNd | GradMatch | TracIn | Data Shapley | DVRL | VTrust-R |
|---|---|---|---|---|---|---|
| SA | 85.1 | 71.1 | 62.76 | 61.13 | 82.1 | **87.0** |

**Why Sampled Augmentation?** We train models for individual augmentations (viz. brightness, impulse noise etc.) and test on all the corrupted test sets. A sample heatmap in Figure 2 for CIFAR10 depicts the difference in performance across augmentations. Based on this heatmap, we develop an algorithm *Sampled Augmentation (SAug)* that samples images based on how far the current model is from the self-trained augmentation model's performance (diagonal elements) which turns out to be best for any augmentation. We define Sampling Number (*SN*) for augmentation $j$ as a normalised difference between the average RA for aug $j$ ($RA_j$) and the self-trained accuracy. We compare SAug vs UAug in Table 2(with dataset size in brackets) where it can be seen that SAug outperforms UAug by a significant margin thus turning out to be our augmented data for further experiments.

**Robustness and Accuracy comparison (VTruST-R) with baselines:** We compare VTruST-R with the baselines in Table 3 where it can be clearly seen that model using clean datasets (*Clean-ST*) performs abysmally in terms of RA, thus indicating the need of augmentations. In some cases, VTruST-R outperforms AugMax in terms of RA even at fractions < 60% (MNIST 20% , CIFAR10 40%), thus indicating that data-centric approaches help in creating quality training datasets. Additionally, we also experiment with some existing data valuation methods (GraNd (Paul et al., 2021), GradMatch (Killamsetty et al., 2021) , TracIn (Pruthi et al., 2020), Data Shapley (Ghorbani & Zou, 2019), DVRL (Yoon et al., 2020)) on CIFAR10 where we train on 20% subset of the *original unaugmented data* and report the Standard Accuracy (SA) in Table 4. We intentionally refrain from reporting the Robust Accuracy (RA) since it is expected to underperform due to the absence of augmented data as seen from Clean-ST's performance in Table 3. In this case too, VTruST is observed to outperform the considered methods by a reasonable margin.

Figure 2: **Performance of self-trained augmentation models on augmented test sets.**

Table 2: **Comparison of Uniform Augmentation with Sampled Augmentation.**

| Methods | Standard Accuracy | Robust Accuracy |
|---|---|---|
| UAug(260K) MNIST | 99.34 | 97.12 |
| SAug(260K) MNIST | **99.37** (0.04) | **97.31** (0.19) |
| UAug(200K) CIFAR10 | 94.84 | 89.06 |
| SAug(200K) CIFAR10 | **94.9** (0.06) | **90.13** (1.07) |
| UAug(300K) TinyImageNet | 60.92 | 26.87 |
| SAug(300K) TinyImageNet | **62.04** (1.12) | **42.04** (15.87) |

Figure 3: **Performance across fractions of subset for varying weights ($\lambda$).**

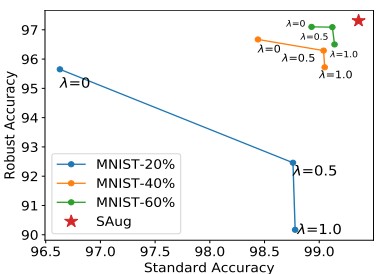

**Tradeoff between Accuracy and Robustness:** We vary the weightage ($\lambda$) on $\mathcal{V}_{rvf}$ and observe in Figure 3 that both the accuracies vary as per $\lambda$, with 20% subset showing considerably larger change. Note that unlike the tradeoff in error-rate/accuracy vs fairness, a large tradeoff is not observed between SA and RA (as also observed by (Yang et al., 2020)) across all the fractions. It is likely to be an artifact of the datasets used.

### 3.3 Data-centric analysis: Post hoc explanation

We pose the following question: *From a data-centric point of view, what characteristics do the selected instances on a per data point level, hold that leads to the fair or robust subset and eventually a fair or robust model?* In this section, we intend to explore some properties/characteristics of the selected samples along with some anecdotal examples.

**Explanation for fairness:**

Exploring the area of counterfactual fairness, we arrive at the question: *Given we change only the sensitive attribute of the selected instances, how does the decision of the model change?* The lesser it changes, fairer the algorithm is. We use the metric *Counterfactual Token Fairness Gap (CF-Gap)*(Garg et al., 2019) and *Prediction Sensitivity*(Maughan & Near, 2020) for our evaluation.

Given a selected instance $x$, we generate a counterfactual instance $x'$ by altering its sensitive attribute. From (Garg et al., 2019), we define CF-Gap$(x)$ as $\|f(x_i) - f(x'_i)\|$ where $f(x)$ corresponds to the model confidence on the target label. Alongside, prediction sensitivity quantifies the extent to which $f(x)$ depends on the sensitive attribute $z$ and is defined as $\frac{\partial f(x)}{\partial z}$. We plot the distribution of both these metrics in Figure 4. It can be observed that VTruST-F acquires the least values across both the measures and datasets, justifying its retainment of fair subsets leading to fair models.

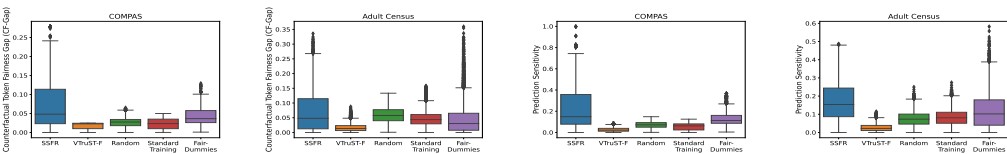

Figure 4: **Box plot representation of CF-gap and prediction sensitivity on the selected subsets from VTruST-F and other baselines.**

We show 10 anecdotal samples from Adult Census dataset in Table 5 on the basis of high *CF-Gap* and we can observe that SSFR has a large number of redundant samples with similar attribute values like that of *(Private, Married-civ-spouse, Husband, White, Male, United States)*, while VTruST-F which anyway has relatively lower CF-gap contains a diverse set of samples. Similar pattern is observed in terms of *Prediction Sensitivity* which we show in the Appendix.

Table 5: **Sample instances with High Counterfactual Token Fairness Gap**

| | | | VTruST-F | | | | | | | | SSFR | | | | |
|---|---|---|---|---|---|---|---|---|---|---|---|---|---|---|
| *Feat* | Age | WCl | MS | Rel | Race | Sex | NC | *Feat* | Age | WCl | MS | Rel | Race | Sex | NC |
| $D_1$ | 25 | Priv | SEP | ORel | B | F | JM | $D_1$ | 55 | Priv | MCS | Husb | W | M | US |
| $D_2$ | 41 | FedG | NM | NIF | W | M | US | $D_2$ | 41 | Priv | MCS | Husb | W | M | US |
| $D_3$ | 43 | StG | NM | NIF | W | M | US | $D_3$ | 29 | Priv | MCS | Husb | W | M | US |
| $D_4$ | 29 | Priv | NM | OC | API | F | TW | $D_4$ | 55 | Priv | MCS | Husb | W | M | US |
| $D_5$ | 60 | Priv | MCS | Husb | W | M | US | $D_5$ | 42 | LoG | MCS | Husb | W | M | US |
| $D_6$ | 35 | Priv | Div | UnM | W | F | US | $D_6$ | 30 | Priv | MCS | Husb | W | M | US |
| $D_7$ | 51 | SEnI | MCS | Wife | W | F | US | $D_7$ | 25 | Priv | NM | NIF | W | M | US |
| $D_8$ | 23 | Priv | NM | OC | W | M | US | $D_8$ | 23 | Priv | NM | OC | W | M | US |
| $D_9$ | 39 | Priv | Div | NIF | AIE | F | Col | $D_9$ | 35 | Priv | MCS | Husb | W | M | DE |
| $D_{10}$ | 34 | Priv | NM | UnM | W | F | DE | $D_{10}$ | 18 | Priv | NM | OC | W | M | US |

**Explanation for robustness:**

Delving into the literature ((Swayamdipta et al., 2020) (Huang et al., 2018)), we pick two measures - *uncertainty* and *distinctiveness*. Having a set of hard-to-learn and distinguishable samples in the subsets makes the model more generalizable and robust. We quantify uncertainty of an instance $x$ in the form of predictive entropy $(-f(x)logf(x))$ and distinctiveness as $\mathbb{E}_{e \in X(s)}dist(fv(x), fv(e))$ where $dist(,)$ is the euclidean distance and $fv(.)$ is the feature from the model's penultimate layer.

We visualize the datapoints in the two dimensions - Uncertainty and Distinctiveness in Figure 5 where we choose a random set of 5000 points from CIFAR and TinyImagenet datasets, followed by marking them as *selected* and *not selected* by VTruST-R and SSR respectively. We can observe that points with relatively high uncertainty and high distinctiveness(HD-HU) values mostly belong to the *selected set of VTruST-R*, while the *unselected points from SSR* mostly cover the HD-HU region.

We also show some anecdotal samples in Figure 6 having High Distinctiveness-High Uncertainty(HD-HU). It can be seen from the anecdotal samples as well as the histogram visual-

isation, that for VTruST-R, diverse samples with difficult augmentations like Impulse noise, Glass Blur are more observed in HD-HU category, while similar(mostly white-background) and unaugmented or No-Noise(NN) samples or easier augmentation based samples like that of brightness are more observed in SSR anecdotal samples, thus justifying the robust selection using VTruST-R.

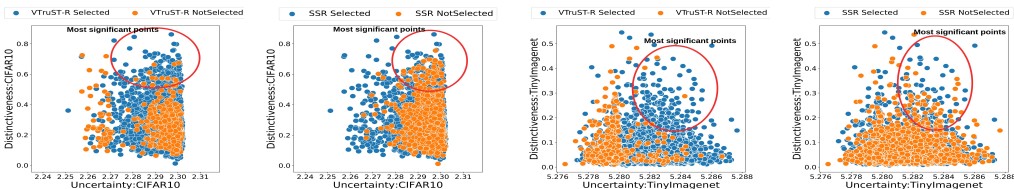

Figure 5: **Data Map for randomly taken 5000 samples from CIFAR10 and TinyImagenet augmented training dataset**

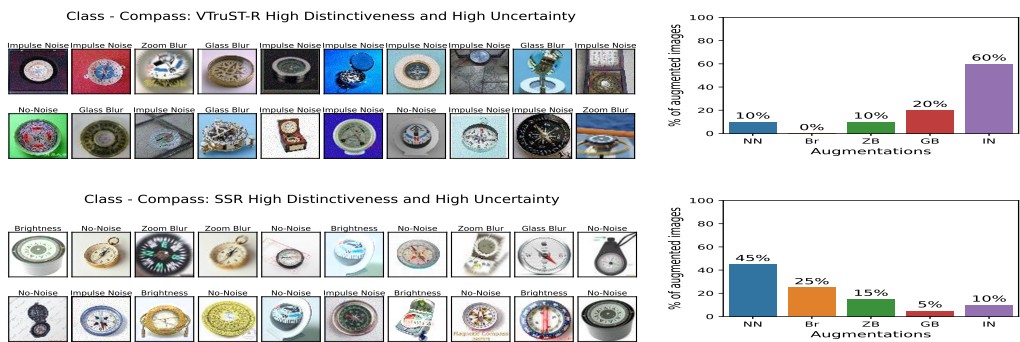

Figure 6: **Anecdotal samples from VTruST-R & SSR with High Distinctiveness and Uncertainty from TinyImagenet for class Compass.**

## 4 RELATED WORK AND DISCUSSION

Besides the usual building of trustworthy models, (Liang et al., 2022) highlights the need for proper data sculpting in order to make the AI systems more trustworthy.The primary task in the robustness and fairness literature is to design robust (Wang et al., 2021), (Chen et al., 2022) and fair (Zemel et al., 2013),(Roh et al., 2020),(Romano et al., 2020),(Sattigeri et al., 2022),(Chuang & Mroueh, 2021) models. To begin with, they use pre-trained models and optimize them further to achieve the desired trustworthy goals. On the contrary, we aim to obtain a subset targeted at a pre-defined trustworthy objective followed by training them from scratch. Broadly, there are two approaches in DCAI for obtaining quality data : (a) data quality measures (Swayamdipta et al., 2020; Ethayarajh et al., 2022; Seedat et al., 2022a;b) and data valuation techniques (Yoon et al., 2020; Ghorbani & Zou, 2019; Pruthi et al., 2020; Das et al., 2021b; Wang & Jia, 2022; Paul et al., 2021). Our work is closely related to the latter among which, most of the works provide a generic method of valuation. The works of (Ghorbani & Zou, 2019; Wang & Jia, 2022) uses value functions that turn out to be expensive due to the usage of MCMC-based approaches. To the best of our knowledge, ours is one of the first works in DCAI that develops a controllable framework to balance the different trustworthiness metrics (fairness and robustness) leading to desired subsets in an online training paradigm. We validate the efficacy of VTruST empirically and also show that this data-centric oriented approach aids in providing data-centric explanations behind its performance.

Exploring suitable value functions for other aspects of trustworthy AI like privacy in the context of subset selection is an open problem. For example, one can use privacy loss as a part of a compositional value function as $\mathcal{V}_p = max(|l(\theta, \mathcal{D}') - l(\theta, \mathcal{D}'_a|)$. The privacy loss shall ensure to retain such datapoints from the training set ($\mathcal{D} \cup \mathcal{D}_a$) that shall keep it at a minimum thus reducing the risk of membership inference attacks thereby maintaining privacy. Hence, one can definitely explore our framework by constructing suitable value functions for the remaining aspects of trustworthy AI.

## REPRODUCIBILITY STATEMENT

We run all our experiments on publicly available datasets and thus all our results can be seamlessly reproduced. We also attach our code (with README) in the Supplementary to aid reproducibility. Details on model architectures and datasets are provided in the main paper. The remaining details for obtaining reproducible results can be found in the Appendix and the attached code repository.

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
