# OpenReview forum: "VTruST : Controllable value function based subset selection for Data-Centric Trustworthy AI"
_ICLR.cc/2024/Conference — Submitted to ICLR 2024_

### Official Review · Reviewer_FQcd · 2023-10-30

**Soundness:** 2 fair
**Presentation:** 3 good
**Contribution:** 3 good
**Rating:** 5
**Confidence:** 3

**Summary:**

This paper studies how to construct high quality datasets for efficient training of trustworthy models. The authors propose a controllable framework for data-centric trustworthy AI, which allow users to control the trade-offs between different trustworthiness metrics of the constructed training datasets. They pose the training data valuation and subset selection problem as an online sparse approximation formulation and propose an online-version of the OMP algorithm for solving this problem.

**Strengths:**

- This paper presents a single unified framework for balancing between the multiple value functions.

- The authors formulate the value functions for multiple metrics including fairness, robustness, and accuracy.

- The proposed framework can provide data-centric explanations behind its performance.

**Weaknesses:**

- The goal of the proposed framework is to obtain a subset of training datapoints suited for trustworthiness metrics. However, the authors aim to augment the datapoints when defining the robust value function. The two objectives appear to be in conflict. The relationship between data selection and data augmentation within the proposed framework for robustness needs further elaboration.

- To ensure trustworthiness, the authors propose to train the model using a subset of the training data points. Does the trustworthy model have lower accuracy compared to the model trained on the original dataset? It would be interesting if the authors could theoretically analyze the impact of the data selection on the accuracy of the trained model.

- In their experiment, the authors evaluate the proposed framework's performance in terms of fairness and robustness independently. How does the framework perform when both fairness and robustness are considered together?

- The efficiency analysis of the proposed framework is missing. The authors aim to construct high quality datasets for efficient training of trustworthy models. However, they neither analyze the framework’s complexity nor provide its running time.

**Questions:**

See above weaknesses.

---

> ### Author Response · Authors · 2023-11-20
>
> We thank the reviewer for their thoughtful feedback and good questions! We address them below.
>
> > The relationship between data selection and data augmentation within the proposed framework for robustness needs further elaboration.
>
> It is a well-known fact that augmentation of training data leads to a robust model [1]. However, it also leads to a larger training dataset with possibly unnecessary datapoints. The data subset selection removes the redundant data points thereby improving the dataset quality and data-centric interpretability as described in the paper. We have also added a discussion for the same under Section 3.2 - second paragraph.
>
> [1] Wang, Haotao, et al. "Augmax: Adversarial composition of random augmentations for robust training." NeurIPS 2021.
>
> > Does the trustworthy model have lower accuracy compared to the model trained on the original dataset?
>
> The trustworthy model can be adapted to have a better accuracy/error rate compared to the model trained on the original dataset. We can vary the parameter $\lambda$ to achieve a balance between accuracy and fairness/robustness. We cite examples of instances where the trustworthy model performs better than the model trained on the original dataset.
>
> Figure 1(b) $\lambda=0.5$ ,  VTrust achieves Label Robust ER = 0.19 and EO Disparity=0.11, which are better than the Wholedata-ST with Label Robust ER = 0.2 and EO Disparity = 0.19.
>
> Figure 1(c) $\lambda=0.5$ , VTrust achieves Feature Robust ER = 0.2 and EO Disparity=0.102, which are better than the Wholedata-ST with Feature Robust ER = 0.213 and EO Disparity = 0.16.
>
>
> > How does the framework perform when both fairness and robustness are considered together?
>
> We have added some results in Section 3.1 for the pareto curves in Figure 1, where we varied the $\lambda$ to show the tradeoff between fairness and robustness (label robust using label flipping and feature robust using SMOTE-generated samples). We can observe that the trend is followed as expected and also performs better than Wholedata-ST in terms of all pairwise metrics. We shall add the results of other baselines in the final version.
>
> > Analyze the framework’s complexity
>
> VTruST has a complexity of $\mathcal{O}(kM(N-k))$ where $k$ = subset size, $M$ = size of the validation set, $N$ = size of training set, $M << N$. We had provided this complexity in Section 2.3 before “Conditions for Optimality of the Selected Subset”.

---

### Official Review · Reviewer_ohGL · 2023-11-01

**Soundness:** 3 good
**Presentation:** 2 fair
**Contribution:** 3 good
**Rating:** 6
**Confidence:** 3

**Summary:**

The authors propose a controllable data selection framework that enables control of the tradeoffs between accuracy and trustworthiness metrics, which are fairness and robustness.
The trustworthiness metrics are incorporated into value functions by measuring the change of the metrics over each epoch and convexly combined with the validation loss. For efficiency, the data selection is done online with the iterative OMP algorithm, where data are chosen in a greedy manner, and the incoming data is decided to be added on the go by comparing with the already selected subset. In the case of the reached budget, incoming data can replace the already chosen one as well only if the incoming data can improve the online sparse approximation value. The authors also provide conditions for the selection of an optimal set of data. This work is said to be the first to attempt the problem of online sparse approximation for data selection. The experiments show good performance for fairness and robustness cases.

**Strengths:**

+ The method is intuitive and makes sense.
+ It considers important trustworthiness metrics for the value function (generalization, fairness and robustness).
+ The authors show good performance of VTruST.

**Weaknesses:**

- No comparison to other more up-to-date (when considering the ICLR submission deadline) data valuation metrics such as Beta Shapley [Kwon, 2021], DAVINZ [Wu, 2022], Data OOB [Kwon, 2023], LAVA [Just, 2023], or CG-score [Nohyun, 2023], which are known to provide more robust values and are efficient to compute.

- The fairness baselines also seem not recent (<= 2021).

- If we are combining different value functions as a convex combination of each, how do they work out together? Are there not cancelling each other's effect?

- The approach for robustness seems to be working only for image datasets.

**Questions:**

- For augmentations that are used, it seems to be quite heuristic. Are there no better ways for robustness?

- How do you apply augmentation to tabular data?

- From the abstract and introduction, it seemed that both fairness and robustness metrics would be combined to achieve both desirable properties, but the main paper only provides the cases when they are separate.

**Minor:**

- Variables are often defined later than introduced, which are hard to trace:

  + what is z0 and z1?
  + and y0 and y1?

---

> ### Author Response · Authors · 2023-11-20
>
> We thank the reviewer for their thoughtful feedback and good questions! We address them below.
>
> Questions:
>
> > Are there no better ways for robustness?
>
> Many notions of robustness (e.g. adversarial robustness, etc.), and their tradeoffs with accuracy have been studied in the literature. We follow the notion of robustness defined in [1], which is not adversarial but based on random perturbations leading to image corruptions. These notions of robustness have also been followed in many subsequent papers, e.g. Augmax [2].
>
> [1] Hendrycks, Dan, and Thomas Dietterich. "Benchmarking neural network robustness to common corruptions and perturbations."ICLR 2019.
>
> [2] Wang, Haotao, et al. "Augmax: Adversarial composition of random augmentations for robust training." NeurIPS 2021.
>
>
> > How do you apply augmentation to tabular data?
>
> In case of tabular data, we showed the tradeoff between Robustness and Fairness using the label-flipping experiment taken from SSFR [3]. However, in the revised document, we have also introduced feature-robustness using SMOTE [4] for augmenting samples.
>
> We show the tradeoff between error rate, fairness, label robustness, and feature robustness in Figure 1 of the revised document. These experiments show the generality of our framework and also motivate the problems of studying tradeoffs between the various trustworthiness metrics.
>
> [3] Roh, Yuji, et al. "Sample selection for fair and robust training." NeurIPS 2021.
>
> [4] Chawla, Nitesh V., et al. "SMOTE: synthetic minority over-sampling technique." Journal of artificial intelligence research 16 (2002).
>
>
>
> > but the main paper only provides the cases when they are separate.
>
> In the revised version, we have provided pairwise tradeoffs between various trustworthiness metrics in Section 3.1, since the pairwise tradeoffs are easy to visualize. However, our method can also be used to improve multiple trustworthiness metrics at the same time, whenever it is possible from the problem point of view. For example,
>
> Figure 1(b) $\lambda=0.5$ , VTrust achieves Label Robust ER = 0.19 and EO Disparity=0.11, which are better than the Wholedata-ST with Label Robust ER = 0.2 and EO Disparity = 0.19.
>
> Figure 1(c) $\lambda=0.5$ , VTrust achieves Feature Robust ER = 0.202 and EO Disparity=0.102, which are better than the Wholedata-ST with Feature Robust ER = 0.213 and EODisparity = 0.16.
>
>
>
>
> ***Weaknesses:*** We would also like to rebut some reviewer comments in the weakness section.
>
>
> > No comparison to other more up-to-date (when considering the ICLR submission deadline) data valuation metrics such as Beta Shapley [Kwon, 2021], DAVINZ [Wu, 2022], Data OOB [Kwon, 2023], LAVA [Just, 2023], or CG-score [Nohyun, 2023], which are known to provide more robust values and are efficient to compute.
>
> The reviewers mention the latest data valuation methods, which we have not compared with. We would like to point out that the methods (from 2023) focus on efficiency and are not specific to the predictive task. On the other hand, our framework is designed to be targeted towards one or more tasks using the value function metrics. Hence, we feel that the comparison with these methods would probably be unfair. However, for completeness, we provide the below comparison with CG-Score. As can be seen, our method outperforms the latest baselines as well in terms of Standard Accuracy (SA) and Robust Accuracy (RA).
>
> ## CG-score vs VTruST on CIFAR10
>
> Subset size |    CG Score       |        VTruST
> ***
> 20% - SA, RA       |   84.11, 77.31    |    **92.25, 85.54**
>
> 40% - SA, RA       |   89.13, 83.25    |    **94.74, 88.23**
>
> 60% - SA, RA       |   92.56, 87.13    |    **94.77, 89.21**
>
>
> > The fairness baselines also seem not recent (<=2021)
>
> We report a comparison with a recent paper -  FairDRO [1] using their metric DCA (Difference of Conditional Accuracy) below. The lower the value, the better is the fairness.
>
> ## COMPAS
>
> Metrics  |    VTruST   |    FairDRO
> ***
>
> ER         |     **0.34**       |     0.43
>
> DCA      |     **0.035**     |     0.04
>
>
> ## Adult Census
>
> Metrics    |    VTruST   |   FairDRO
> ***
>
> ER           |      **0.18**      |      0.21
>
> DCA        |     **0.015**     |      0.02
>
>
> As we can see, our method outperforms this recent baseline in both ER as well as DCA. Since the new method follows a different metric (DCA) compared to the existing baselines (EODisparity), we shall add this result in the appendix of the final version of our paper (due to space constraints).
>
> [1] Jung, Sangwon, et al. "Re-weighting Based Group Fairness Regularization via Classwise Robust Optimization." ICLR 2023
>
> ***Minor:***
>
> We revised it in the updated version.

---

> > ### Comment · Reviewer_ohGL · 2023-11-22
> >
> > I appreciate the authors' response and additional results.
> >
> > Maybe it is unclear to me. However, your robustness notion is defined based on image robustness, then how does it apply to non-image datasets? Also, the augmentations that your provided for tabular datasets are used to address imbalance/fairness not robustness of the features. What are data augmentations you can use to tabular data for improved robustness?
> >
> > Even though the data valuation methods are not explicitly designed to address your defined tasks. However, it does not mean that they cannot be used to address these issues and that their selected subsets will be necessarily worse. CG-score does not utilize the validation data as compared to VTrust, which might be the reason for lower score.
> >
> > Since your method also uses epochs for selecting data, how is that the complexity your provided does not depend on that? Can you please elaborate? Thank you for your time!

---

> > > ### Author Response · Authors · 2023-11-23
> > >
> > > We thank the reviewer for the good questions and insights.
> > >
> > > > Maybe it is unclear to me. However, your robustness notion is defined based on image robustness, then how does it apply to  non-image datasets? Also, the augmentations that your provided for tabular datasets are used to address imbalance/fairness not robustness of the features. What are data augmentations you can use to tabular data for improved robustness?
> > >
> > > We understand the reviewer’s concern about SMOTE being used for data augmentation for robustness. However, note that our framework simply takes the augmented dataset as input and we have already demonstrated the use of label augmentation. So, the actual notion of robustness is only a peripheral concern for the framework.
> > >
> > > > Even though the data valuation methods are not explicitly designed to address your defined tasks. However, it does not mean that they cannot be used to address these issues and that their selected subsets will be necessarily worse. CG-score does not utilize the validation data as compared to VTrust, which might be the reason for the lower score.
> > >
> > > We feel that that asking for us to modify CG-score method to incorporate our value function is a bit unfair since it will involve designing a new algorithm. Note that, CG-score has no concept of utilizing a valuation dataset to define a value function. We have already compared it with the existing method.
> > >
> > > >Since your method also uses epochs for selecting data, how is that the complexity your provided does not depend on that? Can you please elaborate? Thank you for your time!
> > >
> > > We report the per epoch time complexity of all the 3 algorithms. We have updated the text to reflect this. We thank the reviewer for pointing out this mistake.

---

### Official Review · Reviewer_7pAq · 2023-11-02

**Soundness:** 2 fair
**Presentation:** 3 good
**Contribution:** 2 fair
**Rating:** 5
**Confidence:** 3

**Summary:**

The paper proposes a data-centric approach called VTruST for achieving trustworthy AI. It introduces a controllable data valuation framework with multiple trustworthiness metrics. The framework allows users to control the balance between these trustworthiness metrics. The problem of training data valuation and selection is formulated as an online sparse approximation problem, which is solved using a novel online version of OMP. The proposed method outperforms state-of-the-art baselines regarding fairness and robustness by using subsets of the original dataset.

**Strengths:**

1. The paper provides a unified framework for balancing between multiple trustworthiness metrics.
2. The proposed method achieves better performance in terms of fairness and robustness compared to the baselines.
3. The online sparse approximation algorithm efficiently selects subsets of training data based on the specified value functions.
4. The paper includes theoretical derivations of the conditions used in the proposed method.

**Weaknesses:**

1. The value functions of "value" and "robustness" are the same, except that one is the original data and the other is the augmented data. Why are these two metrics that require a trade-off?
2. Here, you only control the weights of multiple objectives through $\lambda$, and users can only adjust the weights. When users need to reset the weights, does your algorithm need to be rerun?
3. Why didn't you compare with other method baselines in your robustness experiments? Additionally, in your experimental results, the two metrics do not have a trade-off relationship. Is it reasonable to set the objectives this way?

**Questions:**

See Weaknesses.

---

> ### Author Response · Authors · 2023-11-20
>
> We thank the reviewer for their thoughtful feedback and good questions! We address them below.
>
>
> > Why are these two metrics that require a trade-off?
>
> Many notions of robustness (e.g. adversarial robustness, etc.), and their tradeoffs with accuracy have been studied in the literature. We follow the notion of robustness defined in Hendryks et a.  [1], which is not adversarial but based on random perturbations leading to image corruptions. We use the same technique to generate the augmented dataset $\mathcal{D}_a$. The value functions are defined as loss on validation datasets. Hence, they are different for accuracy (“value” according to reviewer) which uses $\mathcal{D}’$ and “robustness” which uses $\mathcal{D}'_a$ (see Section 2.2). These value functions guide the selection of important training datapoints, which we find are different from each other. Hence, we report the tradeoff between these two value functions.
>
> [1] Hendrycks, Dan, and Thomas Dietterich. "Benchmarking neural network robustness to common corruptions and perturbations."ICLR 2019.
>
> > When users need to reset the weights, does your algorithm need to be rerun?
>
> Yes, the current algorithm needs to be re-run for different values of $\lambda$. However, $\lambda$ only affects the value function and not the features $X_i$ which are dependent on the training run. Hence, for each training run, we can perform subset set-selection for a grid of $\lambda$ in parallel, without additional inference/parameter update cost.
>
> > Why didn't you compare with other method baselines in your robustness experiments?
>
> We compared the image robustness algorithm with state-of-the-art AugMax (NeurIPS 2021) and SSR (NeurIPS 2021) in Table 5. We also compared our method with latest data valuation method [2] as a response to the second reviewer’s comment and found ours to outperform the former method. We will be happy to compare with any suggested baselines. Besides, we have added some results for the robustness on tabular data in the revised document.
>
> [2] Nohyun, Ki, Hoyong Choi, and Hye Won Chung. "Data valuation without training of a model." ICLR 2023.
>
> > Compare with other method baselines in your robustness experiments; The two metrics do not have a trade-off relationship
>
> While the metrics reported initially on the CIFAR10 showed a weaker tradeoff relationship, that might be a property of the dataset which already contains many images per class. We have added a figure (in Figure 3 of the revised document), which shows the tradeoff more clearly. We thank the reviewer for the comment. We shall report the tradeoff on TinyImagenet, shortly.

---

### Author Response · Authors · 2023-11-20

We thank all reviewers for their time and feedback, which is valuable to help us improve the paper! We address specific comments in the responses to each reviewer.

**Changes in the revised version of the paper**

We uploaded a revised version of the paper to address some of the issues that were raised. It contains the following changes:


1. We performed certain restructuring of the subsections under Section 3 in order to demonstrate the tradeoffs among the trustworthiness metrics on both types of datasets used in this paper.


  - We created Section 3.1 (by including contents from Section 3.1.1) addressing Error rate, Fairness, and Robustness on Tabular Data, and Section 3.2 (by including contents from Section 3.2.1) addressing Accuracy and Robustness on Image Data.

  - We removed Section 3.1.1 and Section 3.2.1.


2. We added a small (2nd) paragraph under Section 3.2 explaining the requirement of augmentation of training data points for robustness which leads to the necessity of subset selection.


3. We added Figures 1(c) and 1(d) to show the various tradeoffs on tabular data.


4. We added Figure 3 to show the clear tradeoff between Standard and Robust Accuracy on the MNIST dataset.


5. We added FairMixup (Mroueh, Youssef. "Fair Mixup: Fairness via Interpolation." ICLR. 2021) to Table 1 as another baseline.

6. We have updated the text in section 2.3 paragraph 2 (last 2 sentences) to clarify that the reported time complexity is per-epoch for all the subset selection algorithms.

---

### Meta-Review · Area_Chair_RZyL · 2023-12-08

**Metareview:**

This paper proposes a unified framework for balancing between multiple trustworthiness metrics. Value functions are defined based on the trustworthiness metrics and an online sparse approximation algorithm is developed for data subset selection based on these value functions.

**Strengths**

(1) The reviewers like the contribution of a unified framework for balancing between multiple trustworthiness metrics.

(2) The reviewers also like the consideration of multiple important trustworthiness metrics (i.e., fairness, robustness, and accuracy) in forming the value functions.

(3) The proposed framework yields good fairness and robustness performance (relative to the tested baselines), including data-centric explanations.

**Weaknesses**

All the reviewers have responded in the AC-reviewer discussion phase.

Some of the concerns have been addressed well in the rebuttal (e.g., explanations of the trade-off between accuracy (original data) vs. robustness (augmented data), additional experiments, clarifications, analysis of the computational efficiency, and recent baselines), evaluation of the proposed framework in terms of the fairness and robustness metrics separately).

However, some key concerns remain, as mentioned by the reviewers (including during the AC-reviewer discussion phase):

(1) Even though the authors have shown how the adjustment of lambda impacts the various metrics, a concrete guideline/strategy for determining the value of the weight lambda in a practical way is missing.

(2) There is a lack of theoretical analysis of the impact of the data selection on the accuracy of the trained model.

(3) Reviewer ohGL has made a valid argument about the lack of qualitative and quantitative comparison with the recent related works on data valuation. The authors should include a discussion on the pros and cons relative to such related works on data valuation, including (but not limited to) those mentioned by this reviewer. For example, a key limitation of the proposed work here is the need for a validation dataset which is not necessary in some of the data valuation works like CG-score [Nohyun, 2023] and DAVINZ [Wu, 2022]. That said, such a discussion (in particular, the limitations) is in no way diminishing the contributions of this work.

The authors are encouraged to address the remaining concerns and account for the reviewers' feedback when revising their paper.

**Justification For Why Not Higher Score:**

Some key concerns remain, as discussed in the meta-review. There is also no strong advocate for acceptance of this work.

**Justification For Why Not Lower Score:**

N/A.

---

### Decision · Program_Chairs · 2024-01-16

Reject